# Recombination shapes the diversification of the *wtf* meiotic drivers

**Yan Wang, Hao Xu, Qinliu He, Zhiwei Wu, Zhen Gong*, Guan-Zhu Han***

College of Life Sciences, Nanjing Normal University, Nanjing, China

## eLife Assessment

This **important** study provides one mechanism that can explain the rapid diversification of poison-antidote pairs in fission yeast: recombination between existing pairs. The evidence is largely **solid**, but the study needs to tone down its claims (as it is not shown that the novel poison-antidote can serve as a meiotic driver) and to address small experimental requests. The work is of interest to scientists studying genetic incompatibilities.

**Abstract** Meiotic drivers are selfish genetic elements that distort fair segregation. The *wtf* genes are poison-antidote meiotic drivers that are experiencing rapid diversification in fission yeasts. However, gene duplication alone is insufficient to drive the diversification of *wtf* genes, given the poison encoded by a newly duplicated *wtf* gene can be detoxified by the antidote encoded by the original *wtf* gene. Here, we analyze the evolution of *wtf* genes across 21 strains of *Schizosaccharomyces pombe*. Knocking out each of 25 *wtf* genes in *S. pombe* strain 972h- separately does not attenuate the yeast growth, indicating that the *wtf* genes might be largely neutral to their carriers in asexual life cycle. Interestingly, *wtf* genes underwent recurrent and intricate recombination. As proof of principle, we generate a novel meiotic driver through artificial recombination between *wtf* drivers, and its encoded poison cannot be detoxified by the antidotes encoded by their parental *wtf* genes but can be detoxified by its own antidote. Therefore, we propose that recombination can generate new meiotic drivers and thus shape the diversification of the *wtf* drivers.

**\*For correspondence:**
gongzhen@nnu.edu.cn (ZG);
guanzhu@njnu.edu.cn (G-ZH)

**Competing interest:** The authors declare that no competing interests exist.

## Introduction

During meiosis, the two alleles at a gene locus are separated into gametes, and each gamete has an equal chance of receiving either allele. This fundamental principle of inheritance, known as Mendel's law of segregation (*Abbott and Fairbanks, 2016*), holds across most genetic loci in most sexual species. However, meiotic drivers, a class of selfish genetic elements, subvert fair segregation during gametogenesis and are transmitted to more than one-half (even to all) of the functional gametes produced by a heterozygote (*Sandler and Novitski, 1957*; *Lyttle, 1991*; *Hurst and Werren, 2001*; *Bravo Núñez et al., 2018b*). Meiotic drivers can spread in a population even when they impose fitness costs on their hosts (*Crow, 1991*; *Lindholm et al., 2016*). However, the spread of a meiotic driver can be thwarted by the costs imposed on its carriers or by its genetic suppressors (*Lindholm et al., 2016*).

The fission yeast *wtf* (*with Tf Long Terminal Repeats*) gene family provides an excellent model to study how meiotic drivers act and evolve (*Hu et al., 2017*; *Nuckolls et al., 2017*). Many *wtf* genes are autonomous one-gene poison-antidote meiotic drivers that encode both a spore-killing poison (short isoform) and an antidote to the poison (long isoform) using alternative transcriptional initiation (*Hu et al., 2017*; *Nuckolls et al., 2017*; *Nuckolls et al., 2020a*). To achieve meiotic drive, all spores are exposed to the poison, whereas only those that inherit *wtf* express the antidote and are rescued (*Hu et al., 2017*; *Nuckolls et al., 2017*; *Nuckolls et al., 2020b*). Some other *wtf* genes can

act as drive suppressors (*Bravo Núñez et al., 2018a*; *Bravo Núñez et al., 2020a*). The poison and the antidote differ only in their N-terminal cytosolic tails containing PY (Leu/Pro-Pro-X-Tyr) motifs. PY motif-dependent ubiquitination promotes the transport of the antidote and the poison (physically interacted with the antidote) from the trans-Golgi network to the endosome, thereby preventing toxicity (*Zheng et al., 2023*).

The *wtf* gene family is experiencing rapid diversification: the *Schizosaccharomyces pombe* reference genome encodes 25 *wtf* genes, some of which are pseudogenes. The copy numbers of *wtf* genes vary greatly among different *S. pombe* strains, and frequent nonallelic gene conversion occurs between *wtf* genes (*Hu et al., 2017*; *Nuckolls et al., 2017*; *Eickbush et al., 2019*). However, these findings are based on a limited number of strains, and the pattern and extent of recombination in the *wtf* genes remain to be fully explored. Moreover, *wtf* driver genes are present in the last common ancestor (LCA) of the fission yeasts *S. pombe*, *S. octosporus*, *S. osmophilus*, and *S. cryophilus*, indicating that *wtf* genes have likely maintained the capacity to drive for more than 100 million years (*De Carvalho et al., 2022*). These fission yeast species carry varying numbers of *wtf* genes, ranging from 5 to 83 (*De Carvalho et al., 2022*). Yet, it remains perplexing how *wtf* genes achieved such diversification. On one hand, gene duplication can give birth to new *wtf* gene copies. On the other hand, a newly duplicated *wtf* gene might not drive because the poison produced by the newly duplicated *wtf* gene can be detoxified by the original *wtf* gene. Like newly duplicated genes, the most probable fate of a new *wtf* duplicate is pseudogenization (*Lynch, 2007*; *Innan and Kondrashov, 2010*). Thus, the vast majority of new *wtf* duplicates experience an early exit from the population, most probably never reaching fixation (*Lynch, 2007*; *Innan and Kondrashov, 2010*). To become a new driver, the new *wtf* copy should evolve coupling new poison and new antidote to the new poison through mutations. But the fate-changing mutations are likely to be rare. It follows that gene duplication might be insufficient to drive the diversification of *wtf* genes.

In this study, we analyzed the diversity and evolution of *wtf* genes in the genomes of 21 strains of *S. pombe* that were sequenced using long-read sequencing approaches (*Tusso et al., 2022*). Through knocking out each of 25 *wtf* genes in *S. pombe* laboratory strain 972h-, no significant attenuated growth was observed, indicating *wtf* genes might be not deleterious in the asexual life cycle. We found that recurrent recombination occurred among *wtf* genes. We generated a novel meiotic driver through artificial recombination between *wtf* drivers, and its encoded poison cannot be detoxified by the antidotes encoded by their parental *wtf* genes but can be detoxified by its own antidote. Therefore, we propose that recombination can generate *wtf* driver with new poisons and might shape the diversification of *wtf* genes.

## Results
### Diversity and evolution of *wtf* genes in fission yeasts

First, we analyzed the diversity and distribution of *wtf* genes in fission yeasts. The *S. pombe* reference genome (strain 972h-) encodes a total of 25 *wtf* genes. For these 25 *wtf* genes, the number of exons varies from 3 to 6 (*Figure 1A*; *Bowen et al., 2003*). To investigate the relationship among exons from different *wtf* genes, we grouped these *wtf* exons into clusters based on nucleotide identity of 0.50 (*Figure 1B and C*). The *wtf* exons were grouped into 10 clusters with >2 members, and 12 exons exist as singletons in the similarity network (*Figure 1B and C*, *Supplementary file 1a*). Only exon 1 of all the 25 *wtf* genes group together in a cluster, indicating that the first exons are well conserved among the *wtf* genes. No other exon is conserved among all the 25 *wtf* genes. Therefore, the evolution of the *wtf* gene structures appears to be highly dynamic.

We next identified *wtf* genes in 21 strains of *S. pombe* that were sequenced using long-read sequencing approaches (*Supplementary file 1b*; *Tusso et al., 2022*). The copy number of *wtf* genes varies among different *S. pombe* strains, ranging from 24 (strain JB879) to 37 (strain JB1206) (*Figure 1D*). Synteny analyses show that the *wtf* genes are present in 20 genetic loci (*Figure 1D*). Multiple *wtf* genes were present in 13 *wtf* loci. Within 20 *wtf* loci, at least one *wtf* gene is present in all of or nearly all of the 21 *S. pombe* strains, suggesting that these 20 *wtf* loci might have originated before the LCA of the 21 *S. pombe* strains. *wtf* pseudogenes are prevalent in many *wtf* loci among the 21 *S. pombe* strains, indicating frequent pseudogenization occurred in the *wtf* genes. These results indicate that *wtf* copy number variation is prevalent among *S. pombe* strains.

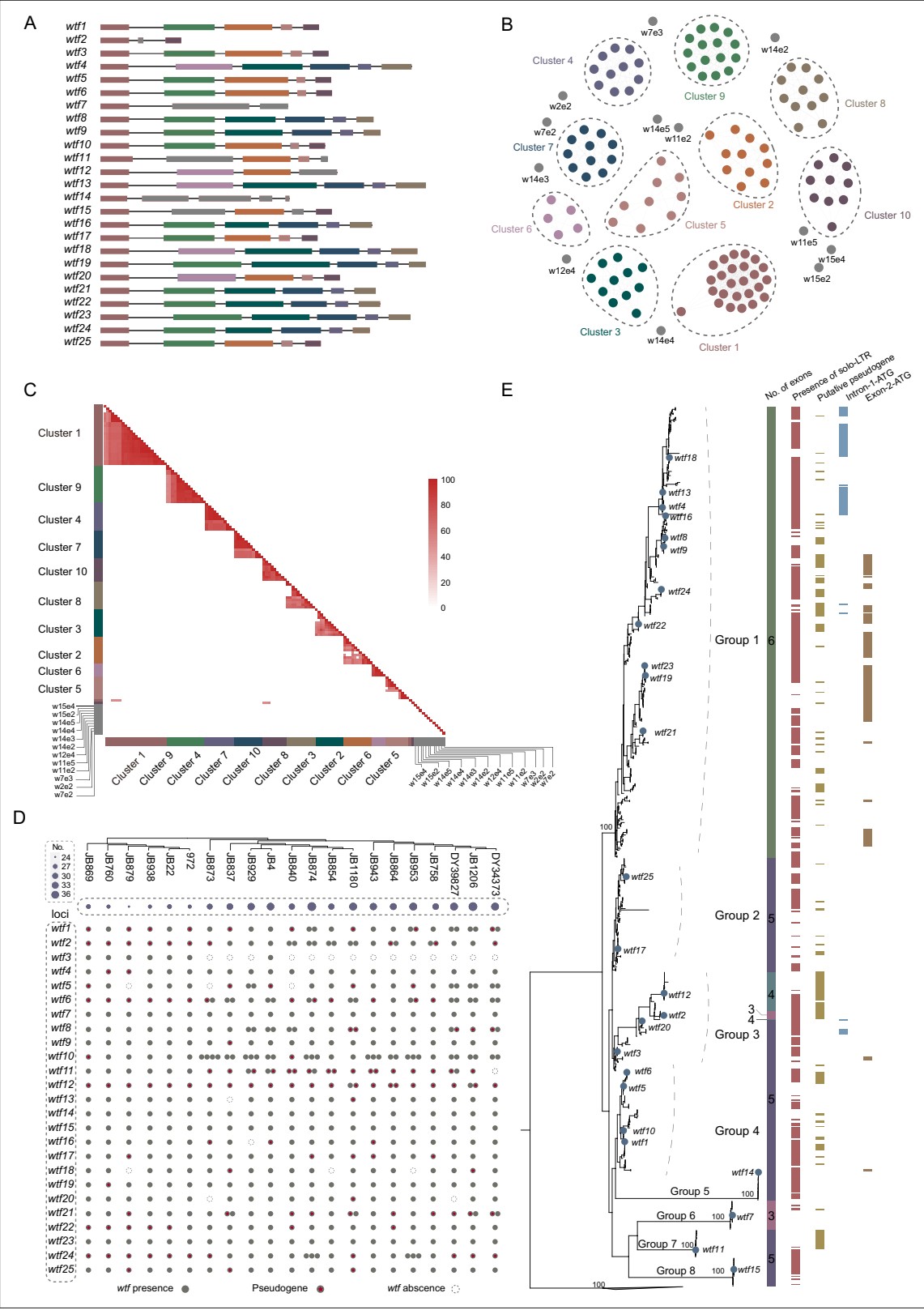

**Figure 1.** The diversity and evolution of *wtf* genes in fission yeasts. (**A**) The gene structures of the *wtf* genes in the *S. pombe* reference genome. Rectangles and lines represent exons and introns, respectively. Rectangles with the same color indicate exons that share >50% sequence identity. (**B**) The similarity network of exons of the *wtf* genes in the *S. pombe* reference genome. Sequences that share an identity of >50% form a cluster. The colors of *wtf* exons correspond to these in panel (**A**). The exons for each cluster were listed in the ***Supplementary file 1a***. (**C**) The heatmap of

*Figure 1 continued on next page*

*Figure 1 continued*

nucleotide identity among exons of the *wtf* genes in *S. pombe* reference genome. (**D**) The distribution of *wtf* genes in 21 *S. pombe* strains. *wtf* genes were present in 25 genetic loci. Filled circles and empty circles represent the presence or absence of *wtf* genes, respectively. Red pentagons indicate putative *wtf* pseudogenes. The size of circles in purple indicates the number of *wtf* genes. The relationship among 21 *S. pombe* strains was inferred based on phylogenetic analysis of 30 randomly selected genes. (**E**) Phylogenetic relationship of *wtf* genes in 21 *S. pombe* strains. The phylogenetic tree is reconstructed using the maximum likelihood method. Filled circles in blue indicate *wtf* genes in *S. pombe* reference genome. The number of exons, the presence of solo-LTRs, the putative pseudogene status, the presence of intron-1-ATG codon, and the presence of exon-2-ATG codon are shown near the corresponding *wtf* gene. *wtf* genes from other fission yeast species were collapsed into a triangle.

The online version of this article includes the following source data and figure supplement(s) for figure 1:

**Source data 1.** Gene structures for wtf1 to wtf25.

**Source data 2.** The sequence identity among all the exons of wtf elements.

**Source data 3.** The distribution and information of wtf genes in 21 *S. pombe* strains.

**Figure supplement 1.** The phylogenetic relationships of solo-LTRs.

We performed phylogenetic analyses of the *wtf* genes from 21 *S. pombe* strains and three other fission yeast species (*S. octosporus*, *S. cryophilus*, and *S. osmophilus*) (*Figure 1E*, *Supplementary file 1c*). The *wtf* genes of *S. pombe* form a monophyletic group. Orthologs of *wtf14*, *wtf7*, *wtf11*, and *wtf15* form monophyletic groups, whereas orthologs of other *wtf* genes show complex phylogenetic mixing, indicating complex recombination might have occurred among these *wtf* genes (*Figure 1E*; *Eickbush et al., 2019*). Moreover, the *wtf* genes with six exons (including the known meiotic drivers *wtf4*, *wtf9*, *wtf13*, and *wtf23*) (*Nuckolls et al., 2017*; *Bravo Núñez et al., 2018a*; *Bravo Núñez et al., 2020a*) cluster together and exhibit a ladder-like phylogeny, which might be generated by continual selection driven by antidotes (like the ladder-like phylogeny of influenza A viruses H1N1 and H3N2, which is shaped by continual immune selection; *Grenfell et al., 2004*; *Bedford et al., 2011*). Based on phylogenetic relationship, we divided the *wtf* genes of 21 *S. pombe* strains into eight groups, namely groups 1–8, among which groups 5–8 include orthologs of *wtf14*, *wtf7*, *wtf11*, and *wtf15*, respectively (*Figure 1E*). Exon 2 ATG codons (exon-2-ATG) and in-frame ATG within intron 1 and near the start of exon 2 (intron-1-ATG) of *wtf* genes can encode the start of poison protein isoforms (*Hu et al., 2017*). We found that most of exon-2-ATG and intron-1-ATG are present within group 1 *wtf* genes (*Figure 1E*). A majority of the *wtf* genes are flanked by solo-LTRs (*Bowen et al., 2003*; *Figure 1E*). However, the solo-LTRs flanking the *wtf* genes do not cluster together but form many distinct groups, suggesting that solo-LTRs were inserted nearby the *wtf* genes multiple times (*Figure 1—figure supplement 1*). Together, our results reveal the rapid diversification and turnover of *wtf* genes in a single fission yeast species.

## No attenuated growth of fission yeast without *wtf* genes

To explore the effect of *wtf* genes on the fitness of fission yeast, we knocked out each of the 25 *wtf* genes in the *S. pombe* laboratory strain 972h- using a method based on homologous recombination (*Figure 2A*). A total of 25 *wtf* knockout strains (*Δwtf1* to *Δwtf25*) were generated. We used spot assay to evaluate the effect of *wtf* gene knockout on the yeast growth, and no growth defect was observed for all the 25 *wtf* knockout strains (*Figure 2B*). Furthermore, no significant differences were observed in the growth curves between the wild-type and *wtf* knockout strains (*Figure 2C*) or in the maximum growth rates among the wild-type and *wtf* knockout strains. Therefore, our experiment suggests that the *wtf* genes might be largely neutral to the fitness of their carriers in the asexual life cycle at least in normal growth condition.

## Recurrent recombination in *wtf* genes

Given complex phylogenetic mixing observed among *wtf* genes (*Figure 1E*), we tested whether recombination occurred. We detected signals of recombination in the 25 *wtf* genes of the *S. pombe* reference genome (p<0.0001) and in the *wtf* genes of the 21 *S. pombe* strains (p<0.0001) using pairwise homoplasy index (HPI) test. Split network analysis also supports the frequent occurrence of recombination in the 25 *wtf* genes of the *S. pombe* reference genome (*Figure 3A*) and in the *wtf* genes of the 21 strains of *S. pombe* (*Figure 3B*). In contrast, no recombination signal was detected for groups 5–8 using HPI test (p=1 for group 5, p=1 for group 6, p=0.53 for group 7, and p=1 for group

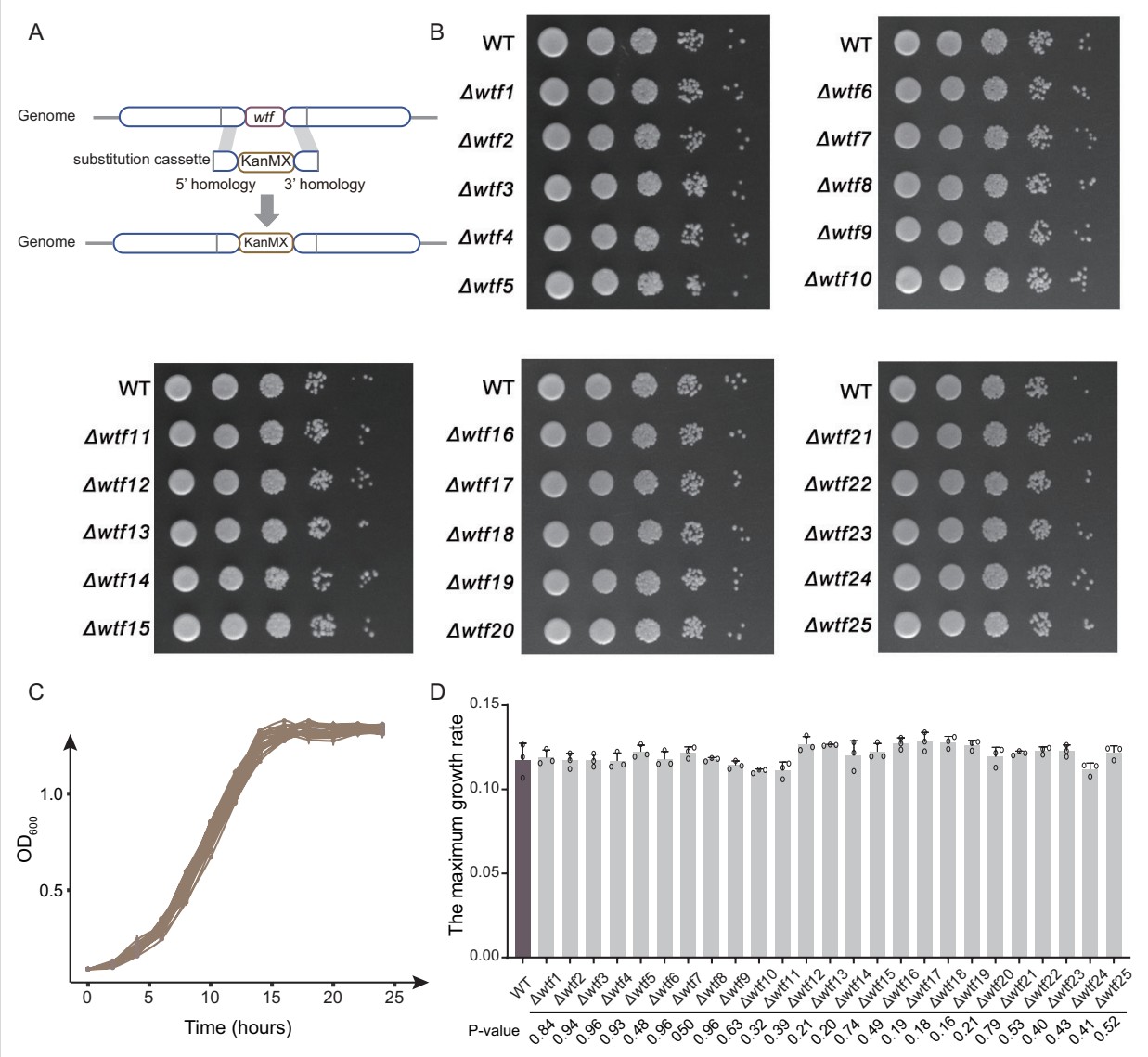

**Figure 2.** The effect of *wtf* gene on the growth of *S. pombe*. (**A**) Generation of *wtf* knockout (*Δwtf*) strains based on the homologous recombination method. Substitution cassette contains a kanMX resistance marker and two homologous sequences flanking the target *wtf* genes. (**B**) Spot assay of *Δwtf* strains. The strains were diluted in five 10-fold steps to $10^{-5}$, and 1.5 μL of each dilution were spotted on the surface of YE solid media. Growth curves (**C**) and maximum growth rates (**D**) of WT and 25 *wtf* knockout strains. Data represent means of three biological replicates (solid lines or bars), with error bars showing SD. Open circles indicate individual replicate values.

The online version of this article includes the following source data for figure 2:

**Source data 1.** Data for growth rates of wtf knockout strains.

8). We estimated recombination rates of the full-length *wtf* sequences, the first exons, and the *wtf* sequences without the first exons for *wtf* groups 1–4. We found that the recombination rate of group 1 *wtf* was highest among the four *wtf* groups (***Figure 3C***). For group 1, breakpoints are dispersed across the *wtf* sequences (***Figure 3D***). These lines of evidence suggest that *wtf* genes underwent recurrent and intricate recombination.

## Generation of a new driver gene through artificial recombination

Given gene duplication alone might be insufficient to shape the diversification of *wtf* genes, we hypothesize that recombination between *wtf* genes can generate new meiotic drives. To test this, we constructed four chimeric *wtf* genes through recombination among known functional meiotic drivers

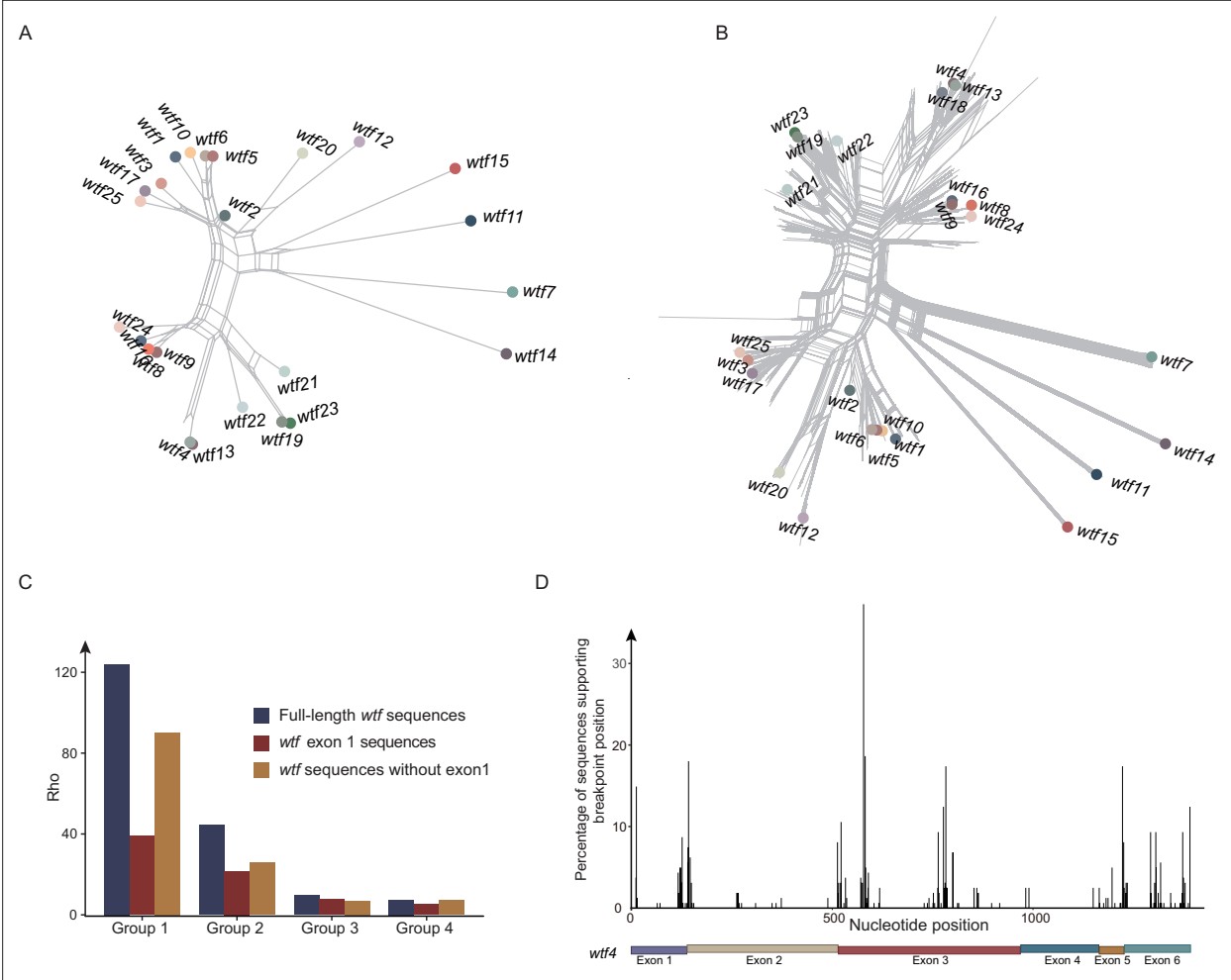

**Figure 3.** Recombination analysis of *wtf* genes. (**A**) Split tree of 25 *wtf* genes of the *S. pombe* reference genome. (**B**) Split tree of *wtf* genes from 21 *S. pombe* strains. The *wtf* genes of the *S. pombe* reference genome are labeled. (**C**) Recombination rates of *wtf* genes that belong to groups 1–4. Recombination rates were estimated for the full-length *wtf* sequences, the first exons, and the *wtf* sequences without the first exons. (**D**) Breakpoints detected for *wtf* genes of group 1. The exons of *wtf4*, as a gene position reference, are shown.

The online version of this article includes the following source data for figure 3:

**Source data 1.** Recombination rates of wtf genes.

(*wtf23*) (*Bravo Núñez et al., 2018a*; *Bravo Núñez et al., 2020a*) and an artificially generated meiotic driver (*wtf18*) as specified below.

We used a proved *Saccharomyces cerevisiae* system to test the activity of poison and antidote proteins encoded by *wtf* genes (*Nuckolls et al., 2020a*). As expected, the expression of the poison proteins (Wtf23$^{poison}$) encoded by *wtf23* genes caused the yeast growth arrest (*Figure 4A*). The attenuated growth was alleviated when the corresponding antidote proteins (Wtf23$^{antidote}$) were expressed (*Figure 4A*). We also experimentally analyzed *wtf18* gene, which was known to encode only long (antidote-like) transcripts and probably act as a suppressor (*Bravo Núñez et al., 2018a*). We artificially introduced an in-frame ATG codon right before the start of exon 2, generating *wtf18$^{poison/-0M}$*. The expression of *wtf18$^{poison/-0M}$* resulted in the yeast growth arrest, suggesting its product, Wtf18$^{poison/-0M}$, is indeed a poison protein (*Figure 4B*). When co-expressing *wtf18$^{antidote}$* and *wtf18$^{poison/-0M}$*, the attenuated yeast growth was rescued (*Figure 4B*), indicating that Wtf18$^{antidote}$ can ameliorate the toxicity of Wtf18$^{poison/-0M}$.

We then constructed four chimeric *wtf* genes through artificial recombination between *wtf23* and *wtf18*, including *wtfC1* (possessing exons 1–2 of *wtf23* and exons 3–6 of *wtf18*), *wtfC2* (possessing exons 1–3 of *wtf23* and exons 4–6 of *wtf18*), *wtfC3* (possessing exons 1–4 of *wtf23* and exons 5–6 of

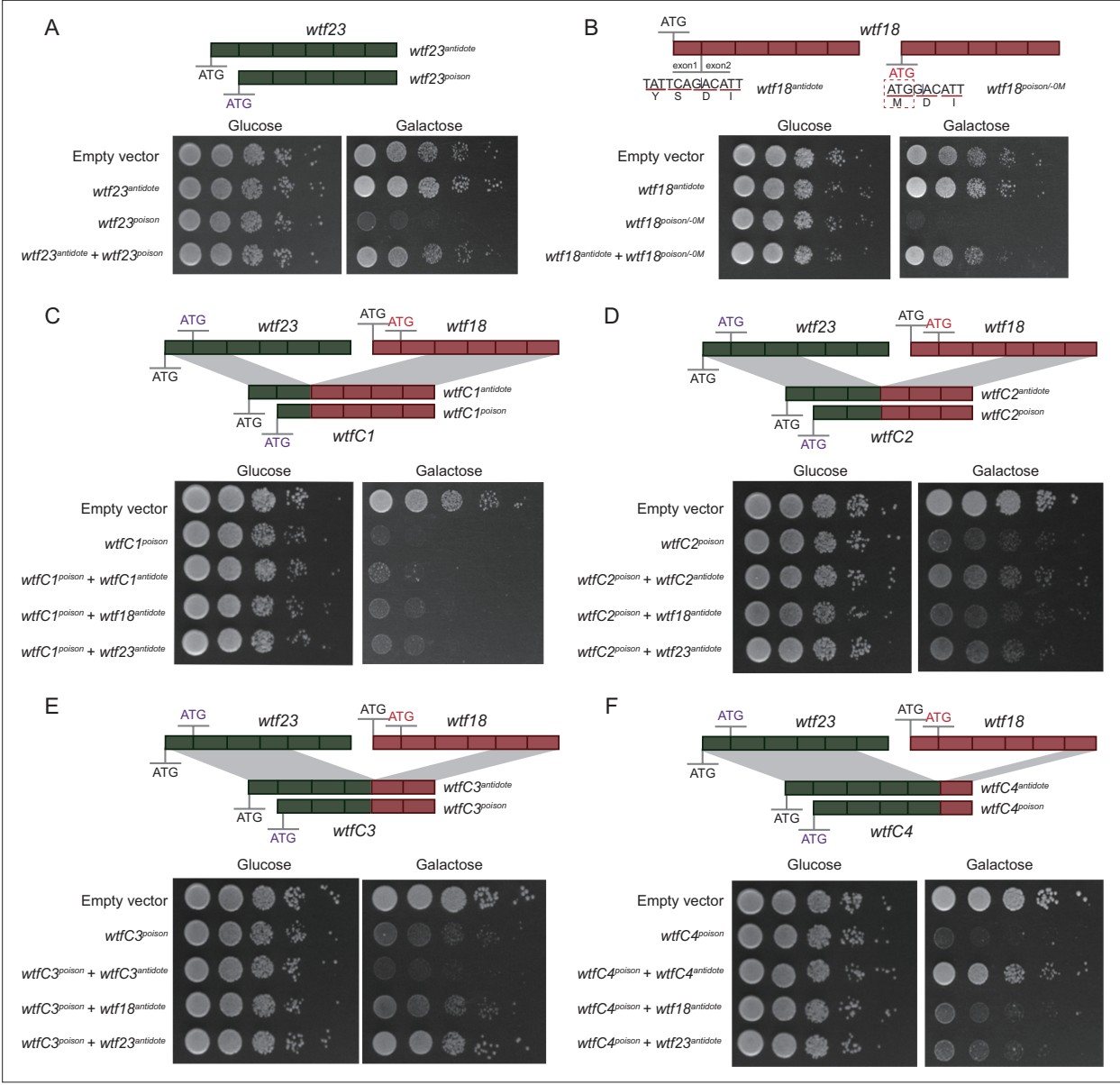

**Figure 4.** Poison and antidote activity of *wtf* genes and chimeric *wtf* genes. *wtf23* and *wtf18* are highlighted in green and red, respectively. Rectangles represent exons. The start codon ATG is shown. For *wtf18*, an in-frame ATG codon was introduced right before the start of exon 2, generating *wtf18^poison/-0M*. Spot assay of yeast transformed with the short isoforms (encoding poison-like proteins) and the long isoforms (encoding antidote-like proteins) of *wtf23* (**A**) and *wtf18* (**B**). (**C–F**) Four chimeric *wtf* genes were generated through artificial recombination, namely *wtfC1* (with exons 1–2 of *wtf23* and exons 3–6 of *wtf18*), *wtfC2* (with exons 1–3 of *wtf23* and exons 4–6 of *wtf18*), *wtfC3* (with exons 1–4 of *wtf23* and exons 5–6 of *wtf18*), and *wtfC4* (with exons 1–5 of *wtf23* and exon 6 of *wtf18*). Spot assay of yeast transformed with the short isoforms (encoding poison-like proteins) and the long isoforms (encoding antidote-like proteins) of *wtfC1* (**C**), *wtfC2* (**D**), *wtfC3* (**E**), and *wtfC4* (**F**) are shown.

*wtf18*), and *wtfC4* (possessing exons 1–5 of *wtf23* and exon 6 of *wtf18*). The expression of the short isoforms of *wtfC1*, *wtfC2*, *wtfC3*, and *wtfC4* resulted in yeast growth arrest, revealing their toxicity (***Figure 4C–F***). However, the antidote of *wtfC1* and *wtfC3* cannot detoxify the corresponding chimeric toxins (***Figure 4C and E***). Interestingly, we generated a putative novel meiotic driver, namely *wtfC4*. Our results show that *wtfC4* encodes a functional poison (WtfC4^poison) (***Figure 4F***). The poison can be detoxified by its own long isoforms (dubbed as *wtfC4^antidote*), but cannot be detoxified by the antidote proteins of their parental genes (***Figure 4F***). Taken together, we generated a new meiotic driver through artificial recombination between pre-existing *wtf* genes.

We tried to test the driver phenotype of *wtfC4* in a more natural setting. We created a recombinant strain, *Sp-wtfC4*, based on the laboratory strain 972h-. Specifically, we replaced the last exon of the original *wtf23* gene with the last exon of *wtf18*. However, we encountered a challenge: since strain 972h- has only one mating type and cannot undergo meiosis on its own, we had to mate the recombinant strain with a BN0 h$^+$ strain that only carries *wtf23$^{antidote}$*. We did not observe the meiotic driver phenotype as expected. This might be due to issues with the proper splicing and expression of the potential poison and antidote proteins or due to the genetic background. Nevertheless, our results raise the possibility that new meiotic drivers can arise through recombination.

## Discussion

In this study, we analyzed the diversity and evolution of *wtf* genes in fission yeasts. The copy number of the *wtf* gene varies among different *S. pombe* strains, revealing rapid diversification and turnover of the *wtf* genes within a single fission yeast species. We detected signals of recurrent and intricate recombination among *wtf* genes as previously reported with limited genomes (*Hu et al., 2017*; *Nuckolls et al., 2017*; *Eickbush et al., 2019*; *De Carvalho et al., 2022*). We hypothesize that recombination between *wtf* genes can produce new *wtf* genes with new poisons and the antidotes to new poisons. These new *wtf* genes can then drive through populations. As proof of principle, we generated a chimeric *wtf* gene that represents a new meiotic driver. The encoded poison of the newly generated meiotic driver can be detoxified by its own long isoforms, but cannot be detoxified by the antidote proteins of their parental genes. However, the other three chimeric *wtf* genes tested did not show this property. Indeed, our recombination breakpoint analyses (*Figure 3D*) reveal substantial

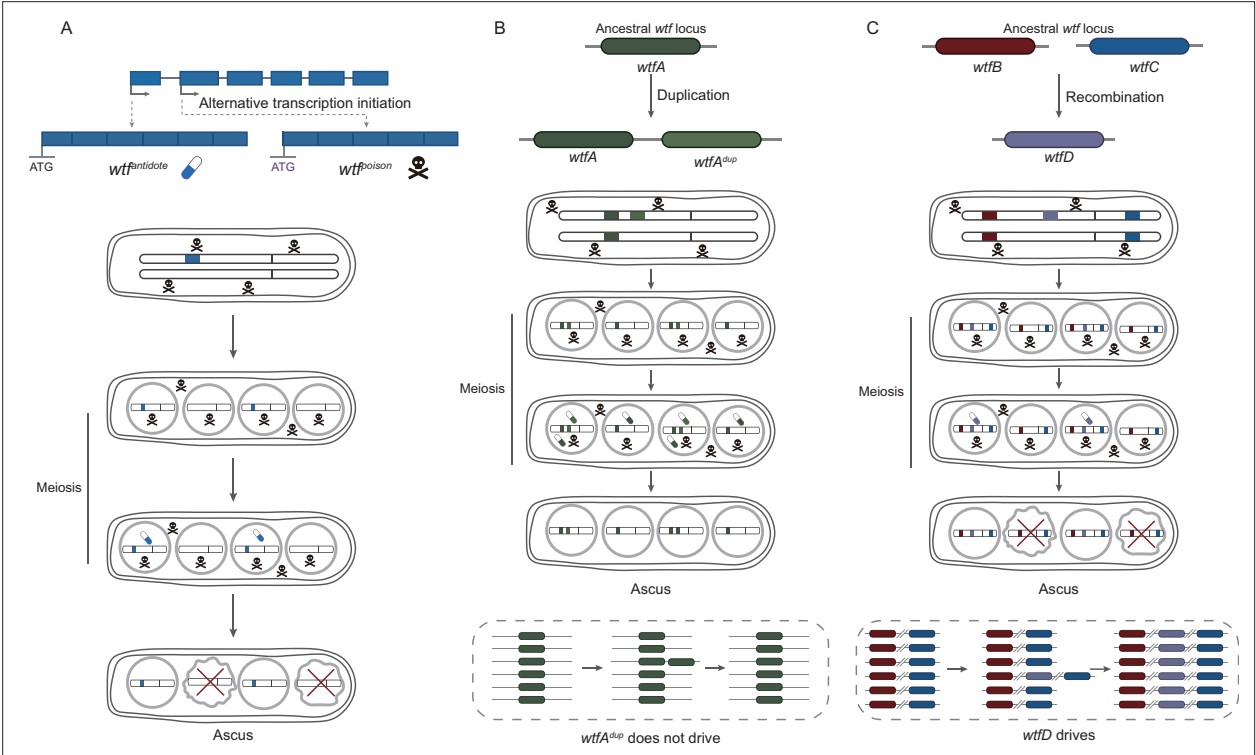

**Figure 5.** The evolutionary mechanisms of *wtf* genes. (**A**) Mechanism of *wtf* meiotic driver. *wtf* genes are transcribed into two transcripts, namely long and short isoforms. The long and short isoforms encode antidote and poison, respectively. All spores are exposed to the poison, whereas only those that inherit *wtf* genes express the antidote and are rescued. Spores without the *wtf* allele are destroyed, resulting in the drive of *wtf*. (**B**) Duplication is insufficient to drive the diversification of *wtf* genes. Gene duplication of *wtfA* gives rise to a new *wtf* gene, *wtfA$^{dup}$*. However, *wtfA$^{dup}$* gene can be detoxified by the original *wtfA* gene and thus cannot drive through its host population. (**C**) When recombination occurs between two parental *wtfB* and *wtfC* genes, generating chimeric *wtfD* gene. *wtfD* encodes a new poison that cannot be detoxified by their parental antidote and the antidote to the new poison. The *wtfD* gene then can spread in its host population.

recombination might have occurred in the last exon. Together, our results indicate that recombination is likely to drive the rapid diversification of *wtf* gene in fission yeasts.

Most of the known meiotic drivers impose costs on their carriers due to direct effects of the driver on survival or fertility, production of a biased sex ratio, or via deleterious mutations linked to the driver (*Price and Wedell, 2008*; *Larracuente and Presgraves, 2012*; *Sutter and Lindholm, 2015*; *Fishman and Kelly, 2015*; *Lindholm et al., 2016*; *Zanders and Unckless, 2019*). In outcrossing between individuals from distinct yeast lineages, *wtf* drivers can provide a selective advantage to atypical spores, such as aneuploids and diploids (*Bravo Núñez et al., 2020b*). In this study, we assessed the effects of the *wtf* genes on the growth of fission yeast during the asexual life cycle through knocking out each of the 25 *wtf* genes in *S. pombe* laboratory strain 972h- separately. We did not observe obvious attenuated growth for these *wtf* knockout strains, indicating *wtf* genes are largely neutral to the fitness of their carriers during the asexual life cycle at least in the normal growth setting. It should be noted that the spot assay used in this study detects only large differences in fitness between wild type and *wtf* knockout strains. Nevertheless, it is likely that *wtf* genes evolve mainly in a neutral manner during the asexual life cycle, which explains the presence of a high proportion of pseudogenes in *wtf* gene repertoire. Moreover, asexual reproduction is much more frequent than sexual reproduction for yeasts (*Tsai et al., 2008*). Therefore, even if fate-changing mutations that simultaneously produce new poison and the antidote to new poison occur, the most probable fate of a new *wtf* gene generated by gene duplication is pseudogenization and removal from the population.

Gene duplication gives rise to new *wtf* genes. However, the newly generated *wtf* gene can be detoxified by the original *wtf* gene and thus cannot drive through its host population when the original *wtf* is fixed in the population (*Figure 5*). Therefore, most, if not all, of the *wtf* gene duplicates experience early exit from the host population. When recombination occurs between two pre-existing *wtf* genes, chimeric *wtf* gene with new poison and the antidote to new poison can be generated as this study shows. Then, the *wtf* gene with new driver property can spread in its host population, even reaching fixation (*Figure 5*). During asexual life cycle, *wtf* genes evolve mainly under genetic drift, and thus can accumulate disruptive mutations, leading to their pseudogenization. Taken together, our study highlights the significance of recombination in shaping the diversification of *wtf* genes.

## Methods
### Identification of the *wtf* genes
We used the blastn algorithm to identify *wtf* genes within 21 *S. pombe* strains with 25 *wtf* genes from *S. pombe* reference genome as the queries and an *e*-cutoff value of $10^{-5}$. The identified *wtf* genes were annotated based on the *wtf* genes of the reference genome. The sequence identity among the exons of *wtf* genes was calculated using BioAider version 1.334 (*Zhou et al., 2020*). Exons were then clustered based on the nucleotide identity using the igraph package version 2.0.1.1 (*Csardi and Nepusz, 2006*; *Csárdi et al., 2024*). We extended 1000 bp flanking each *wtf* gene to establish their syntenic relationships.

### Phylogenetic analysis
The coding sequences of *wtf* genes of 21 *S. pombe* strains and three other fission yeast species (*S. octosporus*, *S. cryophilus*, and *S. osmophilus*) were aligned using MAFFT version 7 (*Katoh and Standley, 2013*). To clarify the relationship of 21 *S. pombe* strains, 30 genes were randomly selected and concatenated using Phylosuite version 1.2.1 (*Zhang et al., 2020*). All the phylogenetic analyses in this study were performed using the maximum likelihood (ML) method implemented in IQ-TREE version 2 (*Minh et al., 2020*). The best-fit substitution model was selected using the ModelFinder algorithm (*Kalyaanamoorthy et al., 2017*). Node supports were assessed using the ultrafast bootstrap approximation (UFBoot) method with 1,000 replicates (*Hoang et al., 2018*). Solo-LTRs were identified using the blast algorithm and aligned using MAFFT version 7 (*Katoh and Standley, 2013*). Phylogenetic analysis was performed using the approximate maximum likelihood method implemented in FastTree version 2.1.1 (*Price et al., 2010*).

## Recombination analysis

Split networks of *wtf* genes were generated using the neighborhood network analysis implemented in SplitsTree4 (*Huson and Bryant, 2006*). Pairwise homoplasy index (PHI) test was performed using SplitsTree4 (*Huson and Bryant, 2006*). Potential breakpoints were detected using 3SEQ (*Lam et al., 2018*). The recombination rate was estimated using the FastEPRR package version 2.0 (*Gao et al., 2016*). The recombination rate was estimated using FastEPRR version 2.0 (*Gao et al., 2016*) as the population-scaled recombination rate, Rho = $4N_e r$, where $N_e$ is the effective population size and r is the per-generation recombination rate. This scaling allows comparison of recombination rates across genomic regions and populations.

## Generation of *wtf* knockout strains

The *wtf* knockout (*Δwtf*) strains generated in this study were derived from *S. pombe* strain 972h-. We constructed substitution cassettes for each of the 25 *wtf* genes of the *S. pombe* reference genome. Substitution cassettes contain a kanMX resistance marker and two homologous sequences flanking the target *wtf* genes (*Moreno et al., 1991*; *García-Ríos et al., 2014*). Substitution cassettes were transformed into *S. pombe* (strain 972h-) through the lithium acetate-based method (*Moreno et al., 1991*; *García-Ríos et al., 2014*). *wtf* gene knockout strains were selected for kanMX resistance and were verified by PCR. The primers used for *wtf* knockout in *S. pombe* are provided in *Supplementary file 1d*.

## Plasmid construction

Total RNA of fission yeast was extracted and reverse transcribed into cDNA. Coding sequences of *wtf23*[antidote], *wtf23*[poison], and *wtf18*[antidote] were amplified using the corresponding primers (*Supplementary file 1e*). *wtf18*[poison/-M0] was generated using *wtf18*[antidote] as the template and the primer with an artificially introduced ATG (*Supplementary file 1e*). We generated *wtfC1* through recombining exons 1–2 of *wtf23* and exons 3–6 of *wtf18*, generated *wtfC2* through recombining exons 1–3 of *wtf23* and exons 4–6 of *wtf18*, generated *wtfC3* through recombining exons 1–4 of *wtf23* and exons 5–6 of *wtf18*, and generated *wtfC4* through recombining exons 1–5 of *wtf23* and exon 6 of *wtf18*. These *wtf* and *wtfC* genes were then cloned into the GAL1/10 dual expression plasmid Gal_HF. Plasmids were first transformed into *Escherichia coli* and verified by PCR and sequencing. Plasmids were then transformed into *S. cerevisiae* (strain S288C) using the lithium acetate-based method. Yeast transformants were selected for kanMX resistance and were verified by PCR.

## Meiotic analysis

We created a recombinant strain, *Sp-wtfC4*, based on the laboratory strain 972h-. Specifically, we replaced the last exon of the original *wtf23* gene with the last exon of *wtf18* using homologous recombination. The *SP-wtfC4* strain and the BN0 h+ strain carrying *wtf23*[antidote] were streaked separately onto YE solid plates and incubated at 30°C for approximately 20 hours. Cells from each strain were then scraped and resuspended in sterile water to an $OD_{600}$ of approximately 0.5, and mixed in equal volumes. The mixture was spread onto SPA plates and incubated upside down at 30°C for 2–3 days to induce meiosis and sporulation. For the fertility assay, 5–10 µL of propidium iodide (PI, 1 mg/mL) was added to 50 µL of $H_2O$, and cells were scraped from the SPA plates and suspended in the PI mix. The mixture was incubated at room temperature for 30 minutes, followed by gentle centrifugation to collect the cells. Fluorescence microscopy was then used for observation and imaging (*Nuckolls et al., 2017*).

## Spot assay

The yeast strains were cultured in YPD liquid medium at 30°C with shaking at 200 rpm. The overnight cultures were transferred to fresh YPD liquid medium and grown to an $OD_{600}$ value of ~3. Cells were collected by centrifugation, and the $OD_{600}$ was adjusted to 3. Subsequently, the strains were diluted in five 10-fold steps to $10^{-5}$, and 1.5 µL of each dilution were spotted on the surface of YPD and YPG solid media. The plates were incubated at 30°C, and the growth of colonies was observed.

## Acknowledgements

This work was supported by the National Natural Science Foundation of China (32270684 and 32470652 to G-ZH and 32300511 to ZG).

## Additional information

### Funding

| Funder | Grant reference number | Author |
|---|---|---|
| National Natural Science Foundation of China | 32270684 | Guan-Zhu Han |
| National Natural Science Foundation of China | 32300511 | Zhen Gong |
| National Natural Science Foundation of China | 32470652 | Guan-Zhu Han |

The funders had no role in study design, data collection and interpretation, or the decision to submit the work for publication.

### Author contributions

Yan Wang, Formal analysis, Investigation, Visualization, Writing – original draft; Hao Xu, Qinliu He, Zhiwei Wu, Investigation; Zhen Gong, Conceptualization, Formal analysis, Supervision; Guan-Zhu Han, Conceptualization, Formal analysis, Supervision, Writing – original draft

### Author ORCIDs

Yan Wang ⓘ https://orcid.org/0009-0002-5375-6082
Guan-Zhu Han ⓘ https://orcid.org/0000-0002-8352-7726

Reviewer #1 (Public review): https://doi.org/10.7554/eLife.100638.3.sa1
Reviewer #3 (Public review): https://doi.org/10.7554/eLife.100638.3.sa2
Author response https://doi.org/10.7554/eLife.100638.3.sa3

## Additional files

### Supplementary files

Supplementary file 1. (a) Exon clusters of *wtf* genes identified in this study. (b) *S. pombe* strains used in this study. (c) *wtf* genes identified in this study. (d) Primer sequences for knockout of the *wtf* genes. (e) Primer sequences used for *wtf* construction.

MDAR checklist

### Data availability

All the data were available in the main text and supplemental information.

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
