## [Editor Report · eLife Assessment]

This **important** study provides one mechanism that can explain the rapid diversification of poison-antidote pairs in fission yeast: recombination between existing pairs. The evidence is largely **solid**, but the study needs to tone down its claims (as it is not shown that the novel poison-antidote can serve as a meiotic driver) and to address small experimental requests. The work is of interest to scientists studying genetic incompatibilities.

---

## [Referee Report · Reviewer #1 (Public review)]

Summary

The authors determine the phylogenetic relation of the roughly two dozen wtf elements of 21 *S. pombe* isolates and show that none of them in the original *S. pombe* are essential for robust mitotic growth. It would be interesting to test their meiotic function by simply crossing each deletion mutant with the parent and analyzing spores for non-Mendelian inheritance. If this has been reported already, that information should be added to the MS. If not, I suggest the authors do these simple experiments and add this information.

Strengths:

The most interesting data (Fig. 4) show that one recombinant (wtfC4) between wtf18 and wtf23 produces in mitotic growth a poison counteracted by its own antidote but not by the parental antidotes. Again, it would be interesting to test this recombinant in a more natural setting - meiosis between it and each of the parents.

Weaknesses:

Some minor rewriting is needed.

Comments on Revision:

(1) The parameter for "maximum growth rate" in Figure 2D needs to be defined and put on the graph.

(2) On page 8, line 182, the authors should consider testing the hybrid wtf in meiosis using strain 975 of Leupold, which is h+, or another standard h+ strain. I don't think the antidote allele is needed; rather, it seems to me it would counter the lethality of the poison protein and should be omitted to test drive of the hybrid wtf. This is a simple experiment and would add considerably to the paper.

---

## [Referee Report · Reviewer #3 (Public review)]

Summary:

In this manuscript, Wang and colleagues explore factors contributing to the diversification of wtf meiotic drivers. wtf genes are autonomous, single-gene poison-antidote meiotic drivers that encode both a spore-killing poison (short isoform) and an antidote to the poison (long isoform) through alternative transcriptional initiation. There are dozens of wtf drivers present in the genomes of various yeast species, yet the evolutionary forces driving their diversification remain largely unknown. This manuscript is written in a straightforward and effective manner, and the analyses and experiments are easy to follow and interpret. While I find the research question interesting and the experiments persuasive, they do not provide any deeper mechanistic understanding of this gene family.

Revision update:

Having read the response to the reviewers, I believe the major issues have been addressed. However, I would strongly suggest toning down the claim regarding the chimeric WTF element in the abstract, which currently reads

"As proof-of-principle, we generate a novel meiotic driver through artificial recombination between wtf drivers, and its encoded poison cannot be detoxified by the antidotes encoded by their parental wtf genes but can be detoxified by its own antidote."

As the author reports in their response, despite various attempts, it was not possible to show that this chimeric WTF element was indeed capable of meiotic drive in a natural context (not transgenic overexpression experiment). thus the authors should not claim they generated "a novel meiotic driver"

Strengths:

(1) The authors present a comprehensive compendium and analysis of the evolutionary relationships among wtf genes across 21 strains of *S. pombe*

(2) The authors found that a synthetic chimeric wtf gene, combining exons 1-5 of wtf23 and exon 6 of wtf18, behaves like a meiotic driver that could only be rescued by the chimeric antidote but neither of the parental antidotes. This is a very interesting observation that could account for their inception and diversification.

Weaknesses:

(1) Deletion strains

The authors separately deleted all 25 Wtf genes in the *S. pombe* ference strain. Next, the authors performed spot assay to evaluate the effect of wtf gene knockout on the yeast growth. They report no difference to the WT and conclude that the wtf genes might be largely neutral to the fitness of their carriers in the asexual life cycle at least in normal growth condition.

The authors could have conducted additional quantitative growth assays in yeast, such as growth curves or competition assays, which would have allowed them to detect subtle fitness effects that cannot be quantified with a spot assay. Furthermore, the authors do not rule out simpler explanations, such as genetic redundancy. This could have been addressed by crossing mutants of closely related paralogs or editing multiple wtf genes in the same genetic background.

Another concern is the lack of detailed information about the 25 knockout strains used in the study. There is no information provided on how these strains were generated or, more importantly, validated. Many of these wtf genes have close paralogs and are flanked by repetitive regions, which could complicate the generation of such deletion strains. As currently presented, these results would be difficult to replicate in other labs due to insufficient methodological details

Revision update:

The authors measured the fitness of the deletion strains using growth curves (Fig. 2C and D) and no significant differences were found, further supporting their claims. The requested information (details on the generation of the deletion strains) is now available in the methods section.

(2) Lack of controls

The authors found that a synthetic chimeric wtf gene, constructed by combining exons 1-5 of wtf23 and exon 6 of wtf18, behaves as a meiotic driver that can be rescued only by its corresponding chimeric antidote, but not by either of the parental antidotes (Figure 4F). In contrast, three other chimeric wtf genes did not display this property (Figure 4C-E). No additional experiments were conducted to explain these differences, and basic control experiments, such as verifying the expression of the chimeric constructs, were not performed to rule out trivial explanations. This should be at the very least discussed. Also, it would have been better to test additional chimeras.

Revision update:

The authors report that the expression of the construct was measured. However, they do not make reference to any specific figure or section of the main text. It would be very useful if the authors explicitly referenced where exactly changes were made (this is true for all changed made)

(3) Statistical analyses

In line 130 the authors state that: "Given complex phylogenetic mixing observed among wtf genes (Figure 1E), we tested whether recombination occurred. We detected signals of recombination in the 25 wtf genes of the *S. pombe* reference genome (p = 0) and in the wtf genes of the 21 *S. pombe* strains (p = 0) using pairwise homoplasy index (HPI) test. "

Reporting a p-value of 0 is not appropriate. Please report exact P-values.

Revision update:

This has been addressed.

---

## [Author Response]

The following is the authors’ response to the original reviews.

**Public Reviews:**

**Reviewer #1 (Public review):**
SummaryThe authors determine the phylogenetic relation of the roughly two dozen wtf elements of 21 *S. pombe* isolates and show that none of them in the original *S. pombe* are essential for robust mitotic growth. It would be interesting to test their meiotic function by simply crossing each deletion mutant with the parent and analyzing spores for non-Mendelian inheritance. If this has been reported already, that information should be added to the manuscript. If not, I suggest the authors do these simple experiments and add this information.

Thanks for the great summary! All the wtf genes have been tested for meiotic drive phenotypes previously by Bravo Nunez et al. (2020; http://doi.org/10.1371/journal.pgen.1008350). The reference was cited in our original manuscript, and we added the details in the revised manuscript.

Strengths:The most interesting data (Figure 4) show that one recombinant (wtfC4) between wtf18 and wtf23 produces in mitotic growth a poison counteracted by its own antidote but not by the parental antidotes. Again, it would be interesting to test this recombinant in a more natural setting - meiosis between it and each of the parents.

Thanks for this insightful comment! As suggested, we have tried to test this recombinant in a more natural setting. We created a recombinant strain (*wtfC4*) based on the laboratory strain 972h-. Specifically, we replaced the last exon of the original *wtf23* gene with the last exon of *wtf18*. However, we encountered a challenge: since strain 972h- has only one mating type and cannot undergo meiosis on its own, we had to mate the recombinant strain with a BN0 h⁺ strain that only carries the *wtf23*^antidote^. Unfortunately, despite of tens of attempts over nearly a year, we did not observe meiotic driver phenotype as expected. This might be due to issues with the proper splicing and expression of the potential poison and antidote proteins or due to the genetic background. Similarly, the drive activity of wtf13 has been shown to be specifically suppressed in certain backgrounds.

Weaknesses:In the opinion of this reviewer, some minor rewriting is needed.

We did the rewriting as this reviewer suggested.

**Reviewer #2 (Public review):**
Summary:This important study provides a mechanism that can explain the rapid diversification of poison-antidote pairs (wtf genes) in fission yeast: recombination between existing genes.

Thanks!

Strengths:The authors analyzed the diversity of wtf in *S. pombe* strains, and found pervasive copy number variations. They further detected signals of recurrent recombination in wtf genes. To address whether recombination can generate novel wtf genes, the authors performed artificial recombination between existing wft genes, and showed that indeed a new wtf can be generated: the poison cannot be detoxified by the antidotes encoded by parental wtf genes but can be detoxified by own antidote.

Thanks for the great summary!

Weaknesses:The study can benefit from demonstrating that the novel poison-antidote constructed by the authors can serve as a meiotic driver.

Thanks for this insightful comment! As suggested, we have tried to test this recombinant in a more natural setting. We created a recombinant strain (*wtfC4*) based on the laboratory strain 972h-. Specifically, we replaced the last exon of the original *wtf23* gene with the last exon of wtf18. However, we encountered a challenge: since strain 972h- has only one mating type and cannot undergo meiosis on its own, we had to mate the recombinant strain with a BN0 h⁺ strain that only carries the *wtf23*^antidote^. Unfortunately, despite of tens of attempts over nearly a year, we did not observe meiotic driver phenotype as expected. This might be due to issues with the proper splicing and expression of the potential poison and antidote proteins or due to the genetic background. Similarly, the drive activity of *wtf13* has been shown to be specifically suppressed in certain backgrounds.

**Reviewer #3 (Public review):**
Summary:In this manuscript, Wang and colleagues explore factors contributing to the diversification of wtf meiotic drivers. wtf genes are autonomous, single-gene poison-antidote meiotic drivers that encode both a spore-killing poison (short isoform) and an antidote to the poison (long isoform) through alternative transcriptional initiation. There are dozens of wtf drivers present in the genomes of various yeast species, yet the evolutionary forces driving their diversification remain largely unknown. This manuscript is written in a straightforward and effective manner, and the analyses and experiments are easy to follow and interpret. While I find the research question interesting and the experiments persuasive, they do not provide any deeper mechanistic understanding of this gene family.

Thanks! Please see the following for our point-to-point response.

Strengths:(1) The authors present a comprehensive compendium and analysis of the evolutionary relationships among wtf genes across 21 strains of *S. pombe*.(2) The authors found that a synthetic chimeric wtf gene, combining exons 1-5 of wtf23 and exon 6 of wtf18, behaves like a meiotic driver that could only be rescued by the chimeric antidote but neither of the parental antidotes. This is a very interesting observation that could account for their inception and diversification.

Thanks for the great summary!

Weaknesses:(1) Deletion strainsThe authors separately deleted all 25 Wtf genes in the *S. pombe* ference strain. Next, the authors performed a spot assay to evaluate the effect of wtf gene knockout on the yeast growth. They report no difference to the WT and conclude that the wtf genes might be largely neutral to the fitness of their carriers in the asexual life cycle at least in normal growth conditions.The authors could have conducted additional quantitative growth assays in yeast, such as growth curves or competition assays, which would have allowed them to detect subtle fitness effects that cannot be quantified with a spot assay. Furthermore, the authors do not rule out simpler explanations, such as genetic redundancy. This could have been addressed by crossing mutants of closely related paralogs or editing multiple wtf genes in the same genetic background.Another concern is the lack of detailed information about the 25 knockout strains used in the study. There is no information provided on how these strains were generated or, more importantly, validated. Many of these wtf genes have close paralogs and are flanked by repetitive regions, which could complicate the generation of such deletion strains. As currently presented, these results would be difficult to replicate in other labs due to insufficient methodological details

We generated growth curves for all the 25 *wtf* deletion strains. We provided the details for *wtf* gene knockout. However, for 25 *wtf* genes, there are too many combinations for editing two genes, and it is technically challenging to knock out multiple *wtf* together. Nevertheless, our results suggest single *wtf* genes have little effect on the host fitness under normal condition.

(2) Lack of controlsThe authors found that a synthetic chimeric wtf gene, constructed by combining exons 1-5 of wtf23 and exon 6 of wtf18, behaves as a meiotic driver that can be rescued only by its corresponding chimeric antidote, but not by either of the parental antidotes (Figure 4F). In contrast, three other chimeric wtf genes did not display this property (Figure 4C-E). No additional experiments were conducted to explain these differences, and basic control experiments, such as verifying the expression of the chimeric constructs, were not performed to rule out trivial explanations. This should be at the very least discussed. Also, it would have been better to test additional chimeras.

We verified the expression of the chimeric genes. The last exon of *wtf18* is too small (128bp) to do more meaningful chimeras.

(3) Statistical analysesIn line 130 the authors state that: "Given complex phylogenetic mixing observed among wtf genes (Figure 1E), we tested whether recombination occurred. We detected signals of recombination in the 25 wtf genes of the *S. pombe* reference genome (p = 0) and in the wtf genes of the 21 *S. pombe* strains (p = 0) using pairwise homoplasy index (HPI) test." Reporting a p-value of 0 is not appropriate. Exact P-values should be reported.

Due to software limitations, the PHI test reports p-values of 0.0 for extremely significant results. We have therefore reported them as <0.0001 in the revised manuscript.

**Recommendations for the authors:**

**Reviewing Editor Comments:**
Regarding the synthetic chimeric wtf gene constructed by combining exons of wtf23 and wtf18, the authors did not explicitly test whether it acts as a meiotic driver in the natural context of a cross. Instead, they examined this possibility only through transgenic overexpression experiments. Given that this is arguably the most important claim of the paper, it is critical that the authors perform, report, and discuss such an experiment in a natural context, regardless of the outcome. It is not necessary to test other recombinants or other wtf loci.

Thanks for this insightful comment! As suggested, we have tried to test this recombinant in a more natural setting. We created a recombinant strain (*wtfC4*) based on the laboratory strain 972h-. Specifically, we replaced the last exon of the original *wtf23* gene with the last exon of *wtf18*. However, we encountered a challenge: since strain 972h- has only one mating type and cannot undergo meiosis on its own, we had to mate the recombinant strain with a BN0 h⁺ strain that only carries the *wtf23*^antidote^. Unfortunately, despite of tens of attempts over nearly a year, we did not observe meiotic driver phenotype as expected. This might be due to issues with the proper splicing and expression of the potential poison and antidote proteins or due to the genetic background. Similarly, the drive activity of *wtf13* has been shown to be specifically suppressed in certain backgrounds.

**Reviewer #1 (Recommendations for the authors):**
The paper is very well written, but some minor points should be corrected or checked.(1) Line 95: Why "Putative"? Is it not clear what a wtf pseudogene is?

“Putative” was removed.

(2) Line 105: Does "known functional" mean they are active (i.e., have been tested and shown to be active)? If so, a reference should be added.

We used “known meiotic divers”, and added reference here.

(3) Line 135: "no recombination signal was tested". Do the authors mean no signal was inferred?

We changed “tested” to “detected”.

(4) Line 147: References for "known functional meiotic drivers (wtf23) and artificially generated meiotic driver (wtf18)" should be given. A statement of how wtf18 was "artificially generated" is essential so the reader knows how that element differs from the wtfC4 generated here.

Reference for *wtf23*. As for *wtf18*, we have specified in the follow text, namely “we artificially introduced an in-frame ATG codon right before the start of exon 2, generating *wtf18poison/-0M*.”

(5) Lines 154 and 424 say an ATG codon was introduced "right before the start of exon 2," but Figure 4B shows it before exon 1.

We thank the reviewer. The introduced ATG is the second start codon in the long transcript and the first in the short transcript. The right panel of Figure 4B shows the short transcript, so the text and figure are consistent.

(6) Line 159: The wtf18 mutant with this additional ATG codon should be tested in meiosis, to see if "putative" is correct.

Thanks. As *wtfC4*, we came with technical challenges to show the driver phenotype in a natural setting, and thus removed this statement.

(7) Line 181: change "driver" to "drive".

Driver is correct.

(8) Line 184: insert to read "wtf genes tested". Also, what is the basis for proposing that "the last exon might be crucial for antidote function"?

“Tested” added, and removed the statement.

(9) Line 198: change to read "detects only large differences".

Done as suggested.

(10) Line 204: change "removed" to "removal".

Done as suggested.

(11) Lines 242 and 243: Are "Splittree4" and "SplitsTree4" different, or is this a misprint?

Corrected!

(12) Lines 274-5 and 412 -3 would read better as "strains were diluted in five 10-fold steps” and “...μL of each dilution spotted on” “…to assay for…"

Done as suggested.

(13) Line 284 says "No new data were generated." This is clearly wrong. Perhaps the authors mean there are no supplementary data files.

Corrected!

(14) Line 406: Change "is" to "are".

Corrected!

(15) Line 413: Surely, they were spotted onto YE agar medium, not liquid medium.

Corrected!

(16) Figure 3C: Define "Rho" and the scale used.

The definition of Rho has been added to the Methods section in the revised manuscript.

**Reviewer #2 (Recommendations for the authors):**
The evidence is largely solid, but the study can benefit from demonstrating that the novel poison-antidote constructed by the authors can serve as a meiotic driver.

As suggested, we have tried to test this recombinant in a more natural setting. We created a recombinant strain (*wtfC4*) based on the laboratory 972h-. Specifically, we replaced the last exon of the original *wtf23* gene with the last exon of *wt18f*. However, we encountered a challenge: since 972h- is a mating-type strain and cannot undergo meiosis on its own, we had to mate the recombinant strain with a BN0 h⁺ strain that carries the *wtf23*^antidote^. Unfortunately, despite of tens of attempts over nearly a year, we did not observe meiotic driver phenotype as expected. This might be due to issues with the proper splicing and expression of the potential poison and antidote proteins.

**Reviewer #3 (Recommendations for the authors):**
I strongly recommend the authors provide all the details concerning the generation of the knock-out strains, including specific primers used (for both the deletion and validation), the result of these validations, and the specific genotype (and ID) of the strains generated.

These details are now included in the Materials and Methods section and in Supplementary.

Please also provide exact P-values (see point 3).

Due to software limitations, the PHI test reports p-values of 0.0 for extremely significant results. We have therefore reported them as <0.0001 in the revised manuscript.